# Febrile Children with Pneumonia Have Higher Nasopharyngeal Bacterial Load Than Other Children with Fever

**DOI:** 10.3390/pathogens12040517

**Published:** 2023-03-27

**Authors:** Bryndis Bjornsdottir, Ubaldo Benitez Hernandez, Asgeir Haraldsson, Valtyr Thors

**Affiliations:** 1Faculty of Medicine, University of Iceland, 101 Reykjavik, Iceland; bryndisbjorns94@gmail.com (B.B.); asgeir@landspitali.is (A.H.); 2Children’s Hospital Iceland, Landspitali University Hospital, 101 Reykjavik, Iceland; 3Department of Science/Biostatistics, Landspitali University Hospital, 101 Reykjavik, Iceland; ubaldob@landspitali.is

**Keywords:** children, respiratory tract infection, pneumonia, bacterial density

## Abstract

Febrile episodes are common in children and the most frequent reason for attending emergency services. Although most infections have a benign and self-limiting course, severe and sometimes life-threatening infections occur. This prospective study describes a cohort of children presenting to a single-centre pediatric emergency department (ED) with suspected invasive bacterial infection, and explores the relationships between nasopharyngeal microbes and outcomes. All children attending the ED who had a blood culture taken were offered to participate over a two-year period. In addition to conventional medical care, a nasopharyngeal swab was obtained., which was analysed for respiratory viruses and three bacterial species using a quantitative PCR. Fisher’s exact test, Wilcoxon rank sum, and multivariable models were used for statistical analyses of the 196 children (75% younger than four years) who were enrolled and had sufficient data for analysis; 92 had severe infections according to the study protocol, while five had bloodstream infections. Radiologically confirmed pneumonia was the most common severe infection found in 44/92 patients. The presence of respiratory viruses and the carriage of *Streptococcus pneumoniae* and *Haemophilus influenzae* were associated with a higher risk of pneumonia. Higher density colonisation with these bacteria were independent risk factors for pneumonia, whereas *Moraxella catarrhalis* carriage was associated with lower risk. Our data support the hypothesis that higher nasopharyngeal density of pneumococci and *H. influenzae* could play a role in the development of bacterial pneumonia in children. A preceding viral infection of the respiratory tract may be a trigger and play a role in the progression to severe lower respiratory tract infection.

## 1. Introduction

Fever is a common symptom in childhood and a frequent reason for consultations at pediatric emergency departments. In most cases, the cause of the fever is an infectious agent. With the widespread use of conjugated vaccines against *Haemophilus influenzae*, meningococci, and pneumococci, the prevalence of bloodstream infections, meningitis, and other invasive bacterial infections has been significantly reduced, and most children presenting with fever have self-limiting infections [1,2,3,4]. It is nevertheless important to understand the pathogenesis of severe infections in order to develop even better preventive measures and more accurate treatment.

The epidemiology of invasive bacterial infections varies by age and geographic regions, but the most prevalently identified causative pathogens in severe pediatric infections are *Staphylococcus aureus, streptococcus pneumoniae*, *Escherichia coli*, Group B streptococci, *Neisseria meningitidis* and *Streptococcus pyogenes* [5,6]. Many of those pathogens are common inhabitants of the skin and mucosal surfaces of the human nasopharynx and gastrointestinal tract. The prevalence and bacterial density of the normal nasopharyngeal colonisers will vary temporally. It may differ geographically and involve the child’s age and socioeconomic status [7]. Pre-school children have the highest prevalence and density of *S. pneumoniae*, *H. influenzae,* and *Moraxella catarrhalis* in the nasopharynx [8,9,10,11]. A large part of the association between age and bacterial density is attributed to the presence of viruses in the respiratory tract [10]. Evidence also suggests that respiratory viral infections will both increase the density of previously colonising bacteria and increase the chance of new bacteria acquisition [12].

A relationship between higher carriage rates of *S. pneumoniae*, *Haemophilus influenzae*, and *M. catarrhalis* and the risk of pediatric pneumonia has been shown in reports from Africa [13] and Asia [14], but data for European children is lacking.

In this study, we prospectively studied children with community-acquired febrile infections with clinical symptoms suggestive of invasive bacterial infections. All enrolled participants had a blood culture drawn. We described their bacterial nasopharyngeal colonisation and density and determined whether they were co-infected with respiratory viruses. Finally, we analysed their outcomes.

## 2. Subjects, Materials, and Methods

### 2.1. Participants and Clinical Definitions

Over a period of 22 months leading up to June 2014, children presenting to the emergency department (ED) at the Children’s Hospital Iceland with symptoms suggestive of severe bacterial infection were prospectively recruited. Children were offered participation if they presented with clinical symptoms suggestive of invasive bacterial infection, and a blood culture was drawn.

Severe bacterial infection was defined as one or more of the following: Bacteraemia, radiographically confirmed pneumonia according to the WHO Pneumococcal Trials Ad Hoc Committee recommendations [15], meningitis, osteomyelitis, septic arthritis, pyelonephritis, and or severe skin/soft tissue infection. A CRP elevation of more than 20 mg/L was defined as suggestive of an infection.

In a subgroup analysis, all children with radiographically confirmed pneumonia were studied concerning nasopharyngeal bacterial carriage and density, and compared with children in the study who had diagnoses other than pneumonia.

All children that met the inclusion criteria, or their parents/caregivers, agreed to take part in the study and gave written consent. The study was approved by the National Bioethics Committee (ref: 12-090) and the Landspítali University hospital’s medical director (ref: 0512-16).

### 2.2. Clinical Data, Sampling, and PCR Analysis

According to the study protocol, nasopharyngeal swabs were obtained from all participants. After collection, the swabs were introduced into a 2 mL container with 1.5 mL skimmed milk-tryptone-glycerol-glucose (STGG) broth [16]. The samples were transported within eight hours of sampling and stored at −80 °C until analysed. For the detection of bacteria and respiratory viruses, a real-time quantitative PCR (qPCR) was used. For the detection of bacteria and determination of bacterial density, the qPCR included pneumococci, *Haemophilus influenzae,* and *Moraxella catarrhalis* using primers previously published [11] (see Appendix A). The viral PCR included adenoviruses, enterovirus, influenza viruses, human metapneumovirus, human parainfluenza viruses, rhinoviruses, and respiratory syncytial viruses using primers previously reported (See Appendix A) [17,18]. The PCR analysis was not a part of the clinical service and did not contribute to the final diagnosis of the patients. Demographic and clinical information, including laboratory values and vital signs at admission to the pediatric ED, were collected.

### 2.3. Statistical Analysis

For data processing and statistical analysis, Microsoft Excel and Stata v.13 were used. Descriptive statistics were applied for all enrolled patients and stratified by pneumonia status (pneumonia vs. non-pneumonia). For continuous variables, comparisons between severe infection status were done using the Wilcoxon rank-sum test. For categorical variables, Fisher’s exact test was used.

To initially assess the association between pneumonia and several explanatory variables, univariable logistic regression analyses were used to obtain crude odds ratios for the explanatory variables, namely age, gender, and presence or absence of viral infection. We calculated contingency tables with the frequencies of pneumonia/non-pneumonia vs. pathogen detected/not detected to visually assess the relationship between infection status and pathogen presence–absence. We formally tested pneumonia status vs. pathogen status with Fisher’s exact test. A logarithmic (log10) transformation was used when calculating bacterial densities and used as a continuous variable in the models. Each variable was used as an explanatory variable in separate models and a pneumonia status outcome. The same variables were then used for an explanatory model using multivariable logistic regression analysis. Diagnostic tests indicated that the model was adequate.

## 3. Results

### 3.1. Demographic Information

At the end of the recruitment period, 225 children had been enrolled, and 196 had sufficient data for analysis and inclusion. Of the 196 children that fulfilled the inclusion criteria, 92 were diagnosed with a severe infection according to the study definitions. Of 196 blood cultures drawn, 10 were positive, of which five (2.6%) grew bacteria defined as pathogenic. To further clarify the approach to data analysis, a flow diagram is shown in Figure 1.

Participants’ demographic information, including age, siblings in pre-school, household smoking, and vaccination status, is shown in Table 1. The mean age of participants was 2 years and 10 months, and the median age was 12 months. Of the participants, 98 (50%) were younger than 12 months, and 147 (75%) were 3 or younger. The number of boys (109, 56%) was similar to girls (87, 44%) (*p* = 0.13). At the time of presentation to the ED, 194 (99%) of the cohort were fully vaccinated according to age, but only 9 (5%) had received the annual influenza vaccine. In total, 41 (21%) children had an underlying health condition, of which asthma was the most common (11, 6%).

### 3.2. Clinical Outcomes and Treatment

An unspecified viral infection was the most common ICD code diagnosis at discharge (n = 53/196), although viral infections were found in more patients. None of these 53 patients were diagnosed with a severe bacterial infection. Of the 92 cases diagnosed with severe infection according to study definitions, pneumonia was the most common infection, especially in 1–3-year-old children (n = 44, Table 2). Seven children had clinical symptoms of sepsis, of whom five had positive blood cultures. Table 2 shows a list of severe infections diagnosed among the children in the cohort, and Figure 2 shows the age distribution for severe infections. Six children (five in the youngest age group) had viral meningoencephalitis, all of whom had a confirmed enteroviral infection.

During their stay at the ED, 121 children (62%) were treated with intravenous antibiotics. The most used empirical antibiotic was ceftriaxone. Of the 121 that received intravenous antibiotics at presentation, 82 (68%) were treated with parenteral antibiotics for more than 48 h, and of those, 50 (61%) were also discharged with prescriptions for oral antibiotics. Of the 196 participants, 116 were admitted for more than 24 h, and the median length of admission was four days. Children with severe infections had longer admission than other children in the cohort (*p* = 0.006). No child needed intensive care admission. The only clinical factors associated with a higher risk of severe infection were oxygen saturation < 95% (*p* = 0.015) and elevated C-reactive protein (*p* = 0.03).

### 3.3. PCR Analysis of Nasopharyngeal Samples

Nasopharyngeal swabs from 94 patients were analysed. (For 102 patients, the sample was missing, DNA extraction was unsuccessful, or the swab was not taken). Of these, 45 were diagnosed with severe infection. The bacterial PCR analysis showed that of the swabs taken from the 94 patients, pneumococci were detected in 30 children (32%), *H. influenzae* in 37 (39%), and *M. catarrhalis* in 45 (48%).

The mean density of pneumococci in positive samples was 2.9 (SD:1.1) log10 gene copies/mL. For *H. influenzae* and *M. catarrhalis,* the densities were 4.2 (SD:1.3) and 4.2 (SD:0.9), respectively.

At least one of the tested viruses was present in 64/94 (68%) of the swabs (See Appendix A). Rhinovirus and respiratory syncytial viruses were the most commonly detected viruses (both n = 17), followed by enterovirus (n = 11). In 43 samples, one viral pathogen was found; in 17 samples, two different viruses were found; and in four samples, three or more viral pathogens were detected.

### 3.4. Relationship between Nasopharyngeal Microorganisms and Pneumonia

We analysed all children diagnosed with pneumonia and compared them with children with other diagnoses regarding nasopharyngeal microorganisms. Of the children who had a nasopharyngeal swab analysed, we found that individuals carrying pneumococci were more likely to have radiologically confirmed pneumonia, and the same was found for the presence of *H. influenzae* (Table 3). However, neither the presence of *M. catarrhalis* nor any type of respiratory virus was associated with the risk of pneumonia. In addition, no associations were observed between the presence of these bacterial species and other forms of severe infection as defined in the study.

In a multivariable analysis of bacterial density, a higher bacterial load as a continuous variable was associated with the risk of radiographically confirmed pneumonia with odds ratios of 1.42 (CI: 1.08–1.88, *p* = 0.012) for *S. pneumoniae* and 1.23 (CI: 1.03–1.47, *p* = 0.012) for *H. influenzae*. The presence of a respiratory virus in the nasopharynx was found to have an adjusted odds ratio for pneumonia of 3.69 (CI: 0.91–14.9, *p* = 0.067). No statistical association was observed between the density of *M. catarrhalis* and the risk of pneumonia, although a mild negative trend was found with an adjusted OR of 0.85 (CI: 0.69–1.05). The complete results of the logistic analysis are shown in Table 4.

The prevalence of other serious infections was too low for meaningful analysis of any contribution of bacterial density.

## 4. Discussion

In this study, we report that a higher density of both pneumococci and *H. influenzae* in the nasopharynx during fever and clinical illness is associated with a risk of radiographically confirmed pneumonia in children. Our results also show an association between the presence of viral pathogens and the risk of radiologically confirmed pneumonia as defined in the study. This supports findings from pediatric studies from Africa and Asia [13,14], but to our knowledge, this is the first study on European children reporting these associations.

The mechanism behind this effect is not known. It may be that viral upper respiratory tract infections lead to higher bacterial density in the nasopharyngeal cavity [10] and subsequent aspiration to the lower respiratory tract. Several studies have described how viral infections can promote changes in the respiratory epithelium [19], mucosal immunity [20], and even changes in the transcriptomic responses of bacterial pathogens [21]. Also, a new acquisition of strains that may proliferate without the effects of the pre-existing microbiome during times of viral upper respiratory tract infections could be of importance [12]. Interestingly, in a report of children with pneumonia in Tanzania, a clear difference was found between bacterial density in four children who died compared to those who survived [13]. Loss of microbiome diversity due to external or internal factors may facilitate bacterial invasion [22], while the nasopharyngeal microbiome may play a direct role in the clinical course following viral URTIs [23]. In addition, although no statistical significance was found, our results suggest that a higher nasopharyngeal density of *M.catarrhalis* may reduce the chance of pneumonia, although the effect is mild. This nevertheless warrants further investigation. Although Moraxella species may provide stability to the nasopharyngeal microbiome and hinder the invasion of more virulent pathogens on its own, it is perhaps more likely that high *M. catarrhalis* density further reflects a favourable composition of bacteria in the nasopharynx or the absence of other pathogenic bacteria [24,25,26]. Children with Moraxella species dominant nasopharyngeal bacterial microbiome were found to have fewer respiratory infections over a two-year period in a study of Dutch infants and young children, which supports our findings [26]. Examples of stable bacterial species protecting against disease have also been suggested in the intestinal microbiome and may reflect similar pathophysiology [27]. The prevalence of nasopharyngeal carriage of the studied bacteria in our study was somewhat lower than many studies have reported [7,28,29]. This may reflect geographic and social differences. The technique of sampling, storing of samples, and analysis may also influence the sensitivity of the PCR.

The exact role of respiratory viruses in the development of bacterial LRTI remains unknown. In our study, there was a strong correlation between pneumonia and the detection of nasopharyngeal respiratory viruses. However, due to small numbers, the confidence intervals were wide, and the effect may be subject to chance.

In our study, elevated CRP levels and SaO_2_ < 95% were associated with a higher risk of serious infection. Neither our study nor other studies [30] have found a single test to rule out invasive infection, and a combination of clinical evaluation and laboratory investigations remain key factors in identifying or ruling out severe infections.

When looking at the demographic information, we see that 75% of the cohort was younger than four years old, probably reflecting that infections in general are more common in this age group. It is important to study this age group better regarding risk and diagnostic factors to minimize hospital admissions and the use of broad-spectrum antibiotics whenever possible. Most of the patients in our study received broad-spectrum intravenous antibiotics at presentation, and many of them received continuous treatment with parenteral antibiotics for more than two days. Several children were discharged with prescriptions for antibiotics despite a diagnosis of a non-severe infection. This fact emphasizes the importance of the sensible use of antibiotics and the usefulness of antimicrobial stewardship/policy in pediatric emergency departments and inpatient wards [31]. Maintaining healthy microbiome patterns through sensible antimicrobial use should be a priority with respect to young children.

Although the epidemiology of pediatric respiratory tract infections has changed since the COVID-19 pandemic, this study helps shed light on the pathophysiology of lower respiratory tract infection and the role of the nasopharyngeal microbiome and viral infections.

There are certain limitations to this study. Firstly, the missing data from several patients, both as background data and lack of nasopharyngeal sampling in half of the patients. This is likely due to the work environment in the ED, where quick and efficient patient flow is important, and attention to detail (registration of data, sampling, and ensuring the proper handling of samples) may be lacking at times. No serotype analysis was done on the samples where pneumococci were detected. This could have given further insights as to whether the carriage of certain serotypes is associated with pneumonia. Universal pneumococcal conjugate vaccination was introduced in Iceland in 2011 with a very high uptake [32]. Since only half of the participants in the study had a successful nasopharyngeal sample, our sample size was limited, and strong assumptions regarding the relationship between severe infection and nasopharyngeal bacterial presence and density are difficult to make. In addition, although an association was found between the density of *S.pneumoniae/H.influenzae* and pneumonia, supported by other studies [13,14], a causal relationship has not been established. Also, considerable time has passed since the execution of the study, and the epidemiology of respiratory infections in children has not followed the usual patterns. Changes have been seen with the RSV epidemiology, and in the 2022–23 season, a high burden of influenza in children has been observed. However, many countries have also been observing high rates of serious bacterial pneumonia as a complication of viral respiratory infections. These observations may further support our findings concerning the association of the interaction of viral and bacterial pathogens in the development of pneumonia.

## 5. Conclusions

In conclusion, we report a study of children who presented with signs and symptoms of potentially severe bacterial infections and found that the presence of respiratory viruses in the nasopharynx and higher density of pneumococci and *H. influenzae* were associated with a higher risk of radiologically confirmed pneumonia.

## Figures and Tables

**Figure 1 pathogens-12-00517-f001:**
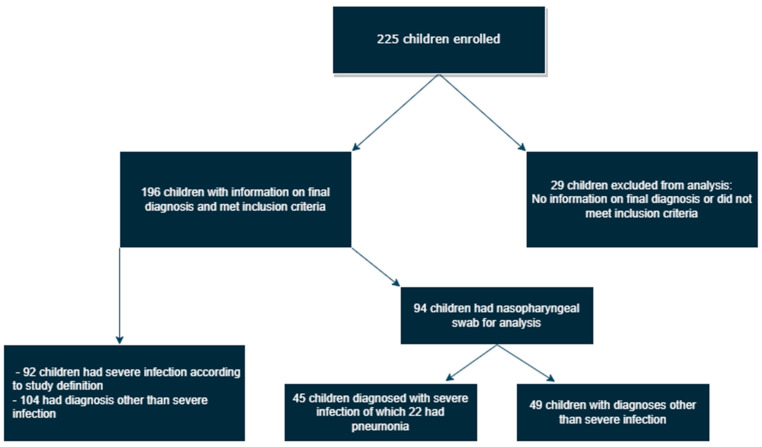
A flow diagram explaining inclusion of patients, categories of infection, and analysis of nasopharyngeal samples.

**Figure 2 pathogens-12-00517-f002:**
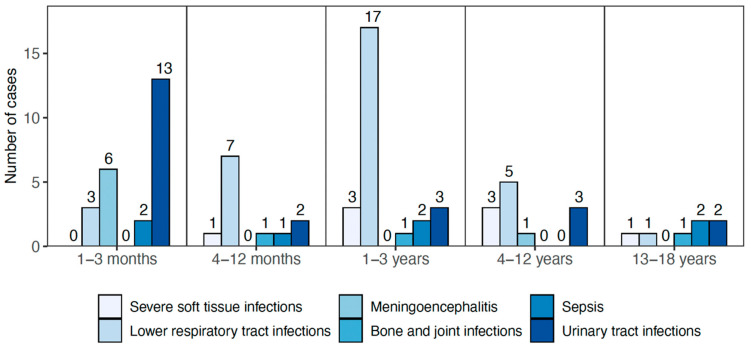
Severe infections, according to the study definition, diagnosed among 196 children at the emergency department at the Children’s Hospital Iceland during a 22-month period, classified by age groups.

**Table 1 pathogens-12-00517-t001:** Demographic information of 196 children who presented to the emergency department at the Children’s Hospital Iceland with symptoms suggestive of a severe infection.

	All Children (n = 196)	Children Diagnosed with Severe Infection (n = 92)
Age		
1–3 months	72 (37%)	43 (32%)
4–11 months	26 (13%)	15 (16%)
1–3 years	49 (25%)	27 (29%)
4–12 years	32 (16, 3%)	14 (15%)
13–18 years	17 (9%)	7 (8%)
Boys	109 (56%)	49 (53%)
Siblings in pre-school	107 (55%)	51 (55%)
Household smoking	36 (18%)	17 (19%)
Vaccination status		
General vaccination schedule up to date	194 (99%)	91 (99%)
Received annual influenza vaccine	9 (5%)	4 (4%)
Any underlying health condition	41 (21%)	24 (26%)

**Table 2 pathogens-12-00517-t002:** Severe infections, according to the study definition, diagnosed in 92 children presenting to the emergency department at the Children’s Hospital Iceland with symptoms suggestive of invasive infections. *: 2 patients had clinical sepsis but negative blood cultures. **: All 6 patients had enteroviral meningoencephalitis.

Severe Infection Type	Number of Cases (% of Severe Infections)
Radiologically confirmed pneumonia [15]	44 (47.8%)
Pyelonephritis	23 (25.0%)
Severe soft tissue infection	9 (9.8%)
Clinical sepsis/bacteraemia *	7 (7.6%)
Meningoencephalitis **	6 (6.5%)
Bone and joint infections	3 (3.3%)

**Table 3 pathogens-12-00517-t003:** Risk of pneumonia in children due to carriage status of bacteria and the presence of any respiratory virus. A total of 22 children with pneumonia and 72 children without pneumonia had a swab analysed.

Nasopharyngeal Swab	Pneumonia n (%)	Non-Pneumonia n (%)	*p*-Value
** *S. pneumoniae* ** **positive**	13/22 (59.1%)	19/72 (26.4%)	0.009
Gene copy number, median (IQR)	352.30 (0–5370)	0 (0–5.42)	0.0013
** *H. influenzae* ** **positive**	15/22 (68.1%)	22/72 (30.6%)	0.002
Gene copy number, median (IQR)	27,905 (0–196,419)	0 (0–357)	0.0001
** *M. catarrhalis* ** **positive**	13/22 (59.1%)	33/72 (45.8%)	0.334
Gene copy number, median (IQR)	430 (0–53,065)	0 (0–12,944)	0.340
**Presence of respiratory virus**	18/22 (81.8%)	46/72 (63.9%)	0.190

**Table 4 pathogens-12-00517-t004:** Risk of radiographically confirmed pneumonia: Results of multivariable logistic analysis of pneumonia status, nasopharyngeal bacterial density, and presence of viruses in 94 children presenting to the emergency department at the Children’s Hospital Iceland with symptoms suggestive of a severe infection. In the logistic model, pneumonia was the outcome; the independent variables were gene copy numbers (log10 transformed) of *S. pneumoniae*, *H. influenzae*, *M. catarrhalis*, presence/absence of viral infection, age, and gender.

Explanatory Variable:	Adjusted Odds Ratio	*p*-Value	95% Confidence Interval
*S. pneumoniae* density (log10)	1.34	0.019	1.05–1.72
*H. influenzae* density (log10)	1.26	0.009	1.06–1.50
*M. catarrhalis* density (log10)	0.85	0.12	0.69–1.05
Presence of respiratory virus	3.69	0.067	0.91–14.9
Age	0.99	0.71	0.98–1.01
Male sex	0.89	0.85	0.28–2.82

## Data Availability

The study data (or parts of it) are available upon request.

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
