# Peer review of "Febrile Children with Pneumonia Have Higher Nasopharyngeal Bacterial Load Than Other Children with Fever"

_pathogens, 2023, doi:10.3390/pathogens12040517_

Round 1

Reviewer 1 Report (Previous Reviewer 2)

Dear authors,

Well done for the effort to change the text, but I think you should try to rewrite the study.

The "problem" of correlation of methods used remains. Also, there is not a flaw in the analysis of the results and is difficult for the reader to understand which patients you include in each group and why, furthermore, how these two groups are correlated with each other.

Moreover, I would agree, the data refer to samples taken in over than a decade, that weakens the whole study.

Maybe you have to enrich your paper with more current data and what I would also suggest is to work under the basement of two groups of patients: patients with severe infection and the results obtained from their swabs and the rest patients and the respective results.

I am sorry to reject the paper.

Author Response

Reviewer 2 Report (New Reviewer)

You reported that they researched on the children who hospitalized due to severe conditions and bacterial road in nasopharynx might related with severity. The theme is interesting for pediatric clinicians. However, there are many unclear points and little points as new knowledge. as follows.

1. The criteria of severe infections is unclear.

2.   You surveyed all children with sever infections, but you described almost only respiratory infections.

3.   The detail data of viral road are not found in your article.

4.  S. pneumoniae and H. influenzae are known as the important pathogens of pediatric respiratory infections. Therefore, the description which bacterial road of them may have relation with severe infections are not thought to be a new know ledge.

5. You should focus on the respiratory infections and contribute another journal.

Author Response

Reviewer 3 Report (New Reviewer)

This prospective study describes a cohort of children presenting to a single centre paediatric emergency department with suspected invasive (bacterial?) infection (disease?) and explores relationships of nasopharyngeal microbes and clinical outcomes.

In short, of n=225 children enrolled into the study the demographic and clinical information was collected from n=196  (aged 1 month to 18 years, median age 12 months, 75% younger than four years) attending the emergency unit of a single hospital in Iceland.  All participants have blood culture taken and a subset of n=94 (48% of 196) has also nasopharyngeal swab collected and tested with qPCR for a panel of respiratory viruses and for three species of respiratory bacteria, namely Streptococcus pneumoniae, Haemophilus influenzae and Moraxella catarrhalis.

Altogether n=92 of 196 participants enrolled have been diagnosed with severe infection according to the “study definition” (or study protocol). Of these 92 children n=44 has radiologically confirmed pneumoniae and five has blood stream (bacterial?) infections.  Study aim was to “explores relationships 

of nasopharyngeal microbes and outcomes (in children (with suspected invasive infection)”. Authors report that the presence of respiratory viruses and the carriage pneumococci and H. influenzae were associated with higher risk of pneumonia, and also that higher density of colonization with these two bacterial species were independent risk factors for pneumonia.

Authors conclude that the study results support the hypothesis that higher nasopharyngeal density of pneumococci and H. inluenzaee could play a role in the development of bacterial pneumonia in children. Reviewer is not convinced the study design allows to test this hypothesis properly. It is also possible that the density of these two bacteria rises in children with symptoms.

Could authors clarify what precisely constitutes a study definition for severe infection and for bacterial pneumonia?

Table 1 provides demographic information for all n=196 children. It seems appropriate to rerport (extra column in table 1?) the same parameters for a subset of 92 children actually diagnosed with severe infection in the study.

It is not clear how many of 53 children diagnosed with viral infection (line 130) were among n=92 diagnosed with severe infection?  How many of 94 children with nasopharyngeal samples collected (line 154) were among n=92 diagnosed with severe infection? How many children in the “non-pneumonia group” (third column in Table 3) were diagnosed with severe infection? If this group was dominated by children with severe infection, would it not be appropriate to exclude children without severe infection when comparing pneumonia to non-pneumonia patients?

Consequently, authors should clarify (a flow chart?) how many children classified with non-severe infection, severe infection and (within group of children with severe infection) with pneumonia have nasopharyngeal sample collected and tested for respiratory pathogens, and how many individuals within these groups were positive among for a pathogen.

In the closing sentence of the manuscript authors state: “We also found potential opportunities for improved use of antibiotics in paediatric emergency departments.” This conclusion seems rather void as none of such opportunities is listed in the paper.

Of note, the reviewer has no access to supplementary information.

MINOR POINTS

Abstract:

Line 11. What invasive infection stands for in the manuscript? Is this invasive bacterial disease, or invasive infection of any aetiology (bacterial, virial, fungal, etc)? Text in lines 40 and 57 implies “invasive infection” represents “invasive bacterial infection”.

Was M. catarrhalis carriage associated with overall reduced risk of pneumonia (compare to what?) or with lower risk of pneumonia compared to the risk associated with S. pneumoniae and/or H. influenzae?

Introduction:

Line 56: Should it not be community acquired febrile infection instead of community acquired febrile episode? In other words, can an episode be community acquired?

Results:

Line 114-116: what “positivity” in 10 children represents here? Can culture be “pathogenic”? Rephrase?

Line 179-180: OR of 0.69-1.05 indicates that there was no statistically significant effect of M. catarrhalis density on presence/absence of pneumoniae, and vice versa.  

Authors mention in the Discussion (line 230) on elevated CRP levels and SaO2<95% being associated in their study with higher risk of serious infection. It seems appropriate to report on CRP levels and blood saturation it in the results section first.

Discussion:

Line 197: Does it concern bacterial pneumonia or pneumonia in general?

Lines 211-212: With no statistically significant difference between children with and without pneumonia, there is no evidence for any associations here.

Spelling in lines: 20 - pneumoniae instead of Pneumoniae; in lines 106, 175 - continuous.

Round 2

Reviewer 1 Report (Previous Reviewer 2)

The manuscript was improved.

Demographic information of patients with no swab, I dont't think that are needed to be reported.

Reviewer 3 Report (New Reviewer)

All my concerns and comments have been addressed by authors.

This manuscript is a resubmission of an earlier submission. The following is a list of the peer review reports and author responses from that submission.

Round 1

Reviewer 1 Report

-        There are several typo errors in the manuscript that authors should carefully revise, particularly spaces between words.

-        The headlines “introduction”, “methods”, “results” and “conclusion” should be deleted from the abstract since this is not the format of the journal.

-        The name of pathogens should be written completely in the abstract: “Streptococcus pneumoniae” instead of “S. pneumoniae” for example.

-        Since the title of the work highlights that “high nasopharyngeal bacterial load” may increase risk of pneumonia in children, authors should clearly state in the abstract how they defined high (versus low) bacterial load using quantitative PCR.

-        In the abstract, authors stated that “This is the first European data suggesting that higher nasopharyngeal density of pneumococci and H. influenzae could play a role in the development of bacterial pneumonia”. Is this true? Since it is a very well-known fact that the presence and load of pathogens in the upper respiratory tract is an increased risk for the development of lower respiratory tract infections. Please clarify this point.

-        In the abstract, authors also stated “This study adds to the understanding of the mechanisms of viral and bacterial interactions and their role in progression to severe pneumonia” but they did not explain the “viral and bacterial interactions”. Please clarify this point.

-        Introduction, name of pathogens should be written completely the first time they appear in the text and then the abbreviation can be used. “Streptococcus pneumoniae” instead of “S. pneumoniae” for example.

-        Material and methods: authors stated “The study was approved by the National Bioethics Committee and the Landspítali University hospital’s medical director”. Please explicitly provide the protocol number approval in this section.

-        Materials and methods: please provide details of the primers used for qPCR as a supplementary table.

-        The study was performed between 2013-2014. Several years have pass and the pandemic of Coronavirus greatly influence the dynamics of respiratory infections globally. Do the authors believe that their data is still relevant?

-        Please improve the quality of Figure 1.

TThe authors provided the mean density of bacteria in positive samples, why did not they show also viral mean densities?

Authors should be carful with the statement “the density of M. catarrhalis was negatively associated with risk of pneumonia” both in the abstract and the main manuscript since association was not statistically significant. Then, there is no association.

 The sections: Supplementary Materials, Author Contributions, Funding, Institutional Review Board Statement, Informed Consent Statement, Data Availability Statement, and Conflicts of Interest are empty.

The authors stated that “It may be that viral upper respiratory tract infections lead to higher bacterial density in the nasopharyngeal cavity and subseqent aspiration to the lower respiratory tract” this seems a very naïve conclusion. There are several works that investigated how viral infections can promote bacterial infections through the alteration of the respiratory epithelium, mucosal immunity and even changes in the transcriptomic response of bacterial pathogens. All this basic research should be considered for the analysis of their results.

Reviewer 2 Report

The present work, by Bjornsdottir et al. presents the results of  study of nasopharyngeal bacterial load and risk of pneumonia in children.

The aim of the study seems interesting, but throughout the whole manuscript the correlation of methods used is not very clear, or is not very well presented.

For example, authors refer number of patients with severe infection and analyse the type of infection, then they refer to an other number of patients of whom they had analysed the swabs, but they don't define the correlation between the 2 groups (how many of the patients of whom a swab was taken are in the group of patients with severe infections?). I think that this is the base on they should start work on the results. 

Furthermore, numbers are not the same. They refer 81 patients with severe infection in line 107, then 91 patients with severe infection in line 123 and then 92 patients in Table 2. 

Also, number of patients enrolled in the study is very low (the real number is 94 patients of whom they finally took a swab). I think that it shoulb be better to have more patients for a more complete study.

As a result, I  think that, even though the aim of the study is very interesting and it would be a significant contribution to the field, the whole manuscript needs major revision and rewriting, methods and results should be correlated and presented in an other more clear and analytical way and it should be better to be rejected to its constant format.